# Beyond Disclosure: Rethinking Transparency and Digital Vulnerability under the Digital Services Act

*Asst. Prof. Dr. Elif Beyza Akkanat-Öztürk*[*]

**Abstract**

The Digital Services Act (DSA) introduces a paradigm shift in platform governance, placing transparency at the heart of regulatory efforts. Yet despite its promise to rebalance power asymmetries between users and platforms, this paper argues that the DSA's transparency obligations may create an unintended "transparency paradox," where the excessive volume and technical nature of disclosures risk reinforcing, rather than reducing, digital vulnerability. Drawing on legal theory and empirical insights, this paper critically assesses the DSA's transparency regime, specifically considering users' cognitive constraints, interface design patterns, and informational inequality It argues that formal compliance with transparency norms does not necessarily yield meaningful understanding or foster true user empowerment. Instead, it may inadvertently obscure the structural power dynamics embedded in platform design and data governance. Building on interdisciplinary research, the paper proposes a shift from mere data-dump transparency toward contextual, user-tested, and layered communication strategies. By reframing transparency as a substantive, user-centric principle, this study offers normative and practical recommendations for European Union (EU) digital regulation to better address digitally enhanced power asymmetries and thereby promote democratic information environments.

**Keywords**: Digital Services Act, transparency paradox, digital vulnerability, user empowerment, information asymmetry, EU platform regulation.

## 1. Introduction

The EU's Digital Services Act (DSA) aspires to reshape digital platform regulation through an ambitious set of transparency obligations. By requiring platforms to disclose the logic of recommender systems, content moderation procedures, advertising practices, and systemic risk assessments, the DSA seeks to recalibrate the relationship between users and intermediaries. At its normative core lies the promise that more transparency will empower users, hold platforms accountable, and mitigate power asymmetries in the digital ecosystem.

However, as transparency becomes the dominant legal tool for reining in platform power, emerging scholarship questions whether these obligations fulfil their empowerment function. Instead of reducing asymmetries, mandatory disclosures may overwhelm users with legalistic or opaque information, leaving them no better off—or worse, falsely reassured ([2]; [3]; [15]).

[*] Istanbul University, Faculty of Law, Comparative Law Department, İstanbul, Türkiye, eba@istanbul.edu.tr, ORCID-ID: 0000-0003-4856-9733.

This paper was presented at the NeXt-generation Data Governance workshop 2025 (NXDG 2025), co-located with SEMANTiCS'25: International Conference on Semantic Systems, September 3–5, 2025, Vienna, Austria.

In the course of this language editing and sentence refinement process, suggestions generated by ChatGPT-4o and NotebookLM were consulted. These technologies were used in a supportive capacity, without fully adopting or directly incorporating all proposed changes.

This phenomenon has been labelled the "transparency paradox": a regulatory condition in which formal transparency obscures rather than clarifies, and procedural openness masks structural dominance ([1]).

This paradox is empirically observable in national contexts as well. A recent study by the Turkish Competition Authority showed that 81.7% of surveyed users did not understand how free digital services are funded, and 71.5% were unaware they were sharing personal data on such platforms ([14]).

This paper interrogates the DSA's transparency framework through the lens of digital vulnerability, a concept that captures the socio-technical fragilities exacerbated by data-driven platforms ([11]). Drawing from interdisciplinary legal theory, consumer protection law, empirical reports ([4]), and EU jurisprudence, it argues that the DSA's model of transparency needs a substantive overhaul. The goal is not merely to critique the shortcomings of current obligations, but to propose a normative reframing: transparency should be seen not as the end itself, but as a means of communicative justice.

To this end, the paper is structured as follows. Section 2 outlines the normative role of transparency in digital regulation and its evolution in EU law. Section 3 maps the architecture of transparency obligations under the DSA. Section 4 introduces the informational crisis and the transparency paradox as structural challenges. Section 5 expands on the concept of digital vulnerability and its intersection with platform asymmetries. Section 6 contextualizes these issues within broader content governance and algorithmic curation practices. Section 7 presents normative and design-based recommendations for a user-centric transparency model. Finally, Section 8 concludes with reflections on transparency's evolving regulatory role.

## 2. Transparency as a Legal Tool in Platform Governance

Transparency has become a foundational regulatory principle in the EU's digital governance framework. From the General Data Protection Regulation (GDPR) to the Platform-to-Business (P2B) Regulation and now the Digital Services Act (DSA), it is widely presumed that transparency can rebalance asymmetries between platform operators and end users, allowing individuals to understand, challenge, or opt out of harmful digital practices. However, as the regulatory reliance on transparency increases, so too does the risk of conflating disclosure with understanding, and formality with fairness.

At its core, transparency is intended to promote accountability and participation. In public law, it is associated with open government and democratic legitimacy; in private law, particularly in consumer protection and contract law, it underpins doctrines of informed consent and fairness ([2]). Within the DSA, transparency assumes a procedural form: users are to be informed about recommender systems, content moderation logic, systemic risks, and advertising parameters. These disclosures are presumed to foster user empowerment

through informed digital choice. If users remain unaware of how so-called "free" digital services are monetized, the very foundation of consent in data exchanges—namely, an understanding of the transaction—is called into question ([14]).

However, the instrumental role of transparency faces critical theoretical challenges. As Ben-Shahar and Schneider famously argue, "mandatory disclosure is the most common and least effective form of regulation" ([5]). When disclosures are excessively long, overly technical, or poorly timed -a common critique seen with GDPR-mandated privacy notices- they risk becoming performative: a checkbox for legal compliance rather than a vehicle for user understanding or control. Consistently, the Turkish Competition Authority reported that nearly 80% of users never revise their privacy settings after initial selection, a trend that reflects the influence of design nudges on user inertia [14]. In the digital context, the opacity of algorithmic systems, behavioural targeting, and personalization techniques render many transparency measures illusory. Scholars have termed this condition translucency: the appearance of openness without genuine visibility ([6]).

Moreover, the assumption that all users are equally positioned to benefit from transparency fails to account for structural inequalities. As *Mišćenić* notes, digital environments create a dual asymmetry: not only do users lack bargaining power, but they are also cognitively and informationally disadvantaged ([2]). This leads to what Liu calls digitally enhanced power asymmetries—platforms leverage scale, opacity, and data-driven insights to deepen their dominance, while users are left with disclosures, they cannot meaningfully process ([3]).

Findings from Türkiye reinforce this diagnosis: over 70% of users believe their personal data is not used for its declared purpose, while more than half express concern over unauthorized access or resale [14].

Thus, although transparency retains normative appeal, its deployment in digital regulation demands closer scrutiny. It is not enough to disclose: the substance, structure, and timing of transparency matter profoundly. The DSA presents an opportunity to rethink transparency not as an end, but as a communicative, contextual, and user-sensitive legal obligation. This rethinking is essential to bridge the widening gap between formal transparency and actual empowerment.

## 3. The Transparency Architecture of the DSA

The DSA introduces one of the most comprehensive transparency regimes in global platform regulation. Its design reflects a legislative ambition to impose visibility on previously opaque systems of algorithmic decision-making, content moderation, and systemic risk management. At the heart of this architecture lies the assumption that disclosure of internal processes will empower users, foster public accountability, and enable regulatory oversight.

Among the core transparency obligations are the publication of content moderation reports (DSA art. 15), the disclosure of the logic behind recommender systems (DSA art. 27), clear notice and justification of content removals (DSA art. 17), and the obligation to perform risk assessments and audits, especially for Very Large Online Platforms (VLOPs) (DSA art. 34-37). Furthermore, the DSA mandates access to ad repositories and transparency around targeting parameters and revenue sources (DSA art. 39). These measures are designed not only for end-users but also for regulators, researchers, and civil society actors.

Yet this ambitious framework is already revealing practical and conceptual shortcomings. Early implementation experiences suggest that platforms often respond to obligations by issuing voluminous, generic, or highly technical reports that, while formally compliant, fail to deliver real insight ([7]). Algorithmic transparency, for instance, is frequently reduced to the disclosure of abstract design principles or conditional logic, offering little substantive information about how personalized feeds are curated or how data inputs shape outputs.

Moreover, empirical research shows wide variation in how platforms approach these obligations. While companies like *Meta* or *TikTok* have developed structured, navigable transparency reports, smaller or less resourced platforms tend to offer minimal, often inaccessible information ([7]; [8]). Even among VLOPs, there is no harmonization in format, terminology, or presentation, making comparisons difficult and limiting the utility of transparency for public scrutiny. This regulatory gap has recently been addressed through a dedicated implementing act by the European Commission ([13]).

The fact that nearly half of Turkish users who read privacy notices do not understand them due to length or complexity, a problem persistent even under frameworks like the GDPR, illustrates the gap between formal transparency and functional comprehensibility-a distinction increasingly vital in assessing regulatory efficacy ([14]).

A further complication stems from the legalistic character of these disclosures. As *Mišćenić* observes, information duties in the digital environment often mirror traditional consumer law models—placing the burden on the user to read, interpret, and act upon standardized disclosures ([2]). This ignores the cognitive and behavioural realities of digital interaction, where users operate in fragmented, time-pressured, and interface-optimized environments. The result is often an information dump: legally exhaustive but practically incomprehensible.

Ultimately, the DSA's transparency architecture reflects a logic of formal accountability that may miss its substantive goal. Compliance is assessed in terms of disclosure quantity and procedural execution, rather than effectiveness or user impact. Without clear criteria for accessibility, standardization, or usability, the DSA risks reinforcing the transparency paradox it seeks to resolve.

**4. The Informational Crisis and the Transparency Paradox**

Despite the normative elegance of transparency, digital regulation increasingly suffers from what scholars have termed an informational crisis—a systemic mismatch between the quantity of disclosed information and the cognitive, temporal, and interpretive capacities of users ([1]). This crisis is not incidental but structural, arising from the regulatory tendency to equate transparency with disclosure volume rather than with communicative efficacy.

At the core of this paradox lies an epistemological assumption: that if information is disclosed, it is thereby understood and can be acted upon. Yet as *Sunstein* ([12]) argues, too much information can be as disabling as too little. Legal mandates that result in verbose, technical, or fragmented disclosures often lead users to ignore, misinterpret, or feel overwhelmed by the information provided ([16]). The architecture of such transparency regimes reflects an intention-action gap, wherein regulators intend to empower, but the structure of delivery leads to passivity or disengagement.

*Ben-Shahar and Schneider* ([5]) provide a foundational critique of what they term the "failure of mandated disclosure." Their empirical and doctrinal analysis demonstrates that disclosure regulation frequently overestimates the rationality and attentiveness of average users, especially when interacting with complex contractual, algorithmic, or platform-based environments. In digital contexts, this critique becomes even more acute: the pace, interface design, and asymmetries of information architecture all contribute to what Liu ([3]) calls digitally enhanced power imbalances.

Further insights from the Turkish Competition Authority's empirical study reinforce this argument, highlighting that 71.5% of users were unaware they had shared personal data while using online platforms, and over 81% had no knowledge of how ad-financed services operate ([14]). Furthermore, privacy policies were often unread or not understood due to excessive length and complexity, confirming the persistence of the transparency paradox even when disclosure is formally fulfilled.

These findings are further reinforced by other aspects of the same survey, which reveal that 70.2% of users believe their data is not used in line with its intended purpose, and 55.4% express concerns over unauthorized use or resale of their data. Additionally, 80% of users never revise their privacy settings once selected, with a majority citing the length and complexity of privacy policies as significant barriers to comprehension. These patterns underscore the existence of the "privacy paradox" in Türkiye, where users' stated concerns about data misuse are not reflected in their actual digital behaviours—mirroring trends observed across other jurisdictions.

These power imbalances are not just about the quantity of information withheld, but also about the design of how information is presented and operationalized. Dark patterns, default settings, nudges, and persuasive UI designs all interact with transparency to create what *Mišćenić* ([2]) describes as a gap between digital fairness and digital formality. Users are

nominally informed—via cookie banners, standard terms, recommender disclosures—but their ability to comprehend and act remains structurally constrained.

The result is a transparency paradox: platforms are more transparent than ever in a procedural sense, yet users are more vulnerable than ever in a substantive sense. Visualising this paradox, *Miščenić* highlights how traders' terms and conditions often meet formal EU transparency requirements but fail the test of intelligibility or real-world empowerment ([2]). This is exacerbated by the dynamic nature of digital environments—where terms are unilaterally modified, disclosures are hidden behind hyperlinks, and standardization is absent.

Furthermore, the phenomenon of translucency, as developed by Rossi (2023), captures a critical regulatory pathology: when platforms disclose in such a way that visibility is simulated but opacity remains. Legal language becomes a shield, not a window; data is disclosed, but not explained.

To resolve this paradox, transparency must be redefined. It must shift from a narrow focus on legal compliance to a broader, interdisciplinary understanding of communicative justice. As *Liu* ([3]) and *Crea & De Franceschi* ([11]) argue, only a user-centric, vulnerability-aware model of transparency can close the gap between rights and reality. The DSA offers a valuable, yet under-realized opportunity to move in this direction.

## 5. Digital Vulnerability and Power Asymmetries

The concept of digital vulnerability has emerged as a critical lens through which to reassess traditional assumptions in consumer protection and platform regulation. Unlike the classical notion of vulnerability—rooted in age, education, or economic status—digital vulnerability is situational, systemic, and interface-driven. It captures how users become exposed to harm not solely because of inherent traits, but because of how digital environments are designed, structured, and governed ([11]).

*Michelle Liu's* ([3]) analysis offers a foundational critique of EU law's assumptions about power asymmetries. She argues that EU instruments, including the DSA, often rely on static models of user weakness—typically equating it with being a "consumer" or a "data subject." Yet, in the digital context, these categorizations are insufficient. Platforms actively shape the user experience through behavioural analytics, data-driven nudging, and algorithmic curation ([16]). Power is not merely a matter of information disparity, but of manipulability: users' choices are not only uninformed, but often pre-structured by design ([15]). The perceived sense of being constantly tracked—identified by Turkish users as a primary source of discomfort in online environments—adds a psychological dimension to digital vulnerability that the DSA's procedural obligations currently overlook ([14]).

This structural manipulability manifests in several layers. First, users are embedded in interface logics where agency is undermined through pre-selected defaults, dark patterns, and

lack of exit options ([2]). Second, as *Irina Domurath* ([9]) explains, users suffer from hypo-autonomy—a condition where their formal rights to choose are retained, but meaningful capacity to exercise those rights is eroded. Digital vulnerability thus resides not in individual fragility but in relational disempowerment: users are rendered fragile by architecture and governance, not by nature.

A particularly acute form of digital vulnerability concerns cognitive asymmetry. As noted by *Goanta et al.* ([10]), users cannot grasp the implications of personalized pricing, automated filtering, or content prioritization, not because they are inattentive, but because these systems are intentionally complex. Algorithmic opacity functions as a control mechanism, and transparency obligations—without standardization or explanation—do little to alleviate this. Rather, they further outsource the burden of vigilance to the already-disadvantaged party.

The current legal framework only partially acknowledges these dynamics. While the DSA addresses systemic risk and mandates independent audits for VLOPs, it does not embed digital vulnerability as a guiding legal principle. Nor does it require design-based equity—the notion that user interfaces should be calibrated to prevent exploitation and enable resilience. This omission reflects a broader regulatory lag: legal regimes struggle to adapt to non-linear, feedback-driven environments where vulnerability is generated in real-time and at scale.

*Crea and De Franceschi* ([11]) call for a paradigm shift—one that places digital vulnerability at the centre of private law reform. This requires new doctrinal tools, such as algorithmic fairness tests, proactive interface standards, and dynamic assessments of user exposure. It also implies institutional shifts: regulatory authorities must develop capacity to evaluate not just the legality of disclosures, but their intelligibility and impact.

In sum, digital vulnerability reframes transparency not as a formal gesture of openness, but as a structural condition of participation. The DSA's future success hinges on its ability to internalize this perspective—not merely by refining obligations, but by transforming the underlying logic of user protection from passive disclosure to active empowerment.

## 6. Intersections with News, Algorithmic Filtering, and Content Regulation

Building on the concept of digital vulnerability, the DSA's transparency obligations, while primarily aimed at regulating platform operations, have far-reaching implications beyond consumer protection -particularly in the domain of news distribution and democratic discourse, where they can exacerbate or mitigate user fragilities. As digital platforms become the primary gateway to journalistic content, the modalities of algorithmic curation, ranking, and visibility directly influence the plurality, quality, and accessibility of information available to the public ([8]).

The ACCC-commissioned report, The Impact of Digital Platforms on News and Journalistic Content (2018), underscores this shift: platforms no longer act merely as intermediaries but

as de facto editors. Their recommender systems shape which stories users see, in what order, and how frequently introducing new layers of algorithmic gatekeeping. These curational processes are typically opaque, even though they profoundly impact civic participation, public trust, and media sustainability.

Transparency, in this context, intersects with both epistemic justice and media pluralism. Users are not simply consumers of digital services but democratic subjects who rely on credible information to form opinions and make choices. The opacity of recommender systems and content delivery mechanisms can thus be viewed as a democratic deficit. As the ACCC notes, news consumers are often unaware of the criteria used to prioritize content or of the economic incentives that shape platform–publisher relationships ([8], Ch. 2–3).

Moreover, the lack of standardization in transparency reporting across platforms contributes to fragmented visibility. *Urman and Makhortykh* (2023) document the inconsistency in how major platforms report on their moderation and recommendation practices, making it nearly impossible to assess systemic patterns or compare across services. This incoherence hampers not only user comprehension but also academic and regulatory scrutiny. The result is a transparency regime that discloses without informing and regulates without enabling democratic oversight.

In addition, algorithmic filtering and content personalization often reinforce filter bubbles and echo chambers, though empirical evidence remains mixed ([8], Ch. 2.4). Still, the DSA's focus on recommender transparency—especially for VLOPs—marks an important first step in mitigating these effects. Yet without accompanying measures to translate disclosures into actionable knowledge (e.g., interface labels, user-choice toggles, or plain-language summaries), these obligations remain inert.

The intersection of platform transparency with journalistic content also reveals deeper tensions around informational integrity. As the ACCC highlights, digital monetization models incentivize short, emotionally charged, and viral content—undermining editorial independence and long-form journalism. This economic restructuring of content production, while not directly addressed by the DSA, is reinforced by its narrow conception of transparency as a procedural duty rather than a substantive safeguard of public goods.

In this light, transparency must evolve to serve democratic ends. This involves not only clarifying the mechanics of algorithmic distribution but also foregrounding the societal role of platforms in shaping public discourse. As *Pasquale* ([12]) and others argue, platforms have become information infrastructures with quasi-public responsibilities. Legal frameworks like the DSA must therefore expand their scope—not just to regulate market failures, but to sustain epistemic diversity and democratic resilience.

**7. Toward Meaningful Transparency: Normative, Design, and Institutional Recommendations**

The preceding analysis reveals that the Digital Services Act's transparency framework, while ambitious, remains entangled in formalism and suffers from limited normative depth. To avoid reinforcing the transparency paradox, the EU must move toward a concept of meaningful transparency—one that is not merely procedural but enables actual understanding, decision-making, and agency. This shift entails both legal reform and design-based intervention.

### 7.1. From Disclosure to Communication: Layered and User-Tested Formats

Transparency obligations should be guided not by quantity, but by communicative clarity, leveraging modern IT tools for dynamic and interactive user engagement. As *Ben-Shahar & Schneider* recommend ([5]), disclosures must be tailored to human cognitive limits—employing visuals, summaries, and progressive layers of detail. Rather than static, long legalistic documents, platforms should be required to implement layered and interactive transparency formats: a brief user-friendly explanation followed by expandable technical detail, allowing users to actively explore information tailored to their context.

This approach also demands rigorous usability testing, which can help bridge the intention-action gap observed in the privacy paradox by ensuring information is not only visible but also actionable and empowers users to make meaningful choices. Much like accessibility standards in disability law, information disclosures should be empirically evaluated for intelligibility. The GDPR mandates "clear and plain language" in its articles 12–14 yet offers no systematic mechanism to verify comprehension. The DSA could fill this gap by introducing transparency impact assessments—requiring platforms to demonstrate that disclosures are not just visible but graspable ([2]).

### 7.2. Standardization and Interoperability of Transparency Formats

The lack of harmonized formats undermines the comparative and regulatory utility of transparency obligations. A legally binding EU Transparency Standard—co-developed by the Commission, academic experts, and civil society—could address this issue. Such a standard should specify layout, terminology, data structure, and even colour coding for key disclosures ([7]).

This standardization would also facilitate regulatory benchmarking and civic auditing. Academic institutions, journalists, and NGOs could compare platform behaviour more reliably, fostering external accountability and public trust.

### 7.3. Transparency as Design Governance

Legal obligations should not be limited to content; they must extend to presentation and delivery, enabling dynamic user control and contextual interaction. Transparency must be integrated into user interface (UI) design: opt-out mechanisms should be prominent and

persistent; recommender settings should be explained through interactive prompts, and content moderation criteria should be displayed contextually at the point of action (e.g., when a user is flagged or sanctioned), facilitating informed and dynamic consent.

*Crea & De Franceschi* ([11]) advocate for design-based fairness: legal principles embedded in visual and interactive architecture, not just in policy documents. This would mean regulating transparency not as an isolated duty but as part of broader governance-by-design, aligning with the growing call for human-centric digital environments.

## 7.4. Institutional Guidance and Transparency Metrics

Lastly, regulators must develop tools to assess the effectiveness of transparency. The DSA foresees audits and risk assessments but lacks qualitative transparency metrics—benchmarks for user understanding, behavioural influence, or misinformation resilience. An EU Observatory for Digital Transparency, potentially embedded within the European Board for Digital Services, could centralize evaluations, produce annual reports, and issue interpretive guidelines. Much like the EDPB under the GDPR, such a body would lend coherence to fragmented enforcement and close the gap between formal compliance and real-world impact, ensuring regulatory efforts effectively promote user agency and address issues like the privacy paradox.

## Concluding Remarks: The Early Fallacy?

The Digital Services Act aspires to reshape digital governance through enhanced transparency. Yet, as this paper has shown, the current architecture of transparency risks repeating an early fallacy: the belief that more disclosure equals more empowerment. Despite unprecedented formal openness, digital platforms continue to concentrate power, obscure control, and exploit vulnerability. The paradox is evident: we live in a regime of transparent opacity.

Transparency, as deployed in the DSA, has been conceptualized as a legal remedy. But the structural and behavioural dimensions of digital interaction require us to treat it as a governance mode—one that must be carefully calibrated, empirically evaluated, and ethically grounded. Procedural openness is no substitute for communicative justice.

This paper has argued for a shift toward meaningful transparency: layered, user-tested, and embedded into interface design. It has highlighted the need for standardization, institutional guidance, and a reconceptualization of transparency as a relational duty, not a mere information dump. Legal scholars and regulators alike must resist the tendency to treat transparency as a cure-all, and instead recognize its limits, conditions, and contexts.

Ultimately, the true test of the DSA's success will lie not in how much platforms disclose, but in whether users can understand, challenge, and shape the digital environments they

inhabit. As digital vulnerability continues to evolve, so too must the legal imagination. Transparency may be a starting point—but never the destination.

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
