# OpenReview forum: "Beyond Disclosure: Rethinking Transparency and Digital Vulnerability under the Digital Services Act"
_SEMANTiCS.cc/2025/Workshop/NXDG — NXDG 2025_

### Official Review · ~Rigo_Wenning1 · 2025-07-18
**An old issue with a new perspective**

**Rating:** 6
**Confidence:** 5

**Review:**

The article demonstrates in very long paragraphs the obvious fact that transparency alone does not automatically lead to a better world. We knew that even before cookie-banners. It is interesting to see that most of the very common sources for that argumentation are not cited, that the author comes from a different world. There is no citation of Alessandro Acquisti or Aleecia McDonald (the cost of reading privacy policies). But there are very nice statistics from a market that isn't on the radar of those western scholars. The presentation of the state of the art thus gives new arguments and new perspectives on a known issue. And this is worthwhile presenting!

For the conclusions, they are the classic ones from legal scholars with only a limited understanding of the possibilities of IT tools. Because all that information is overwhelming, there needs to be some standardised language and other reductions of bandwidth. This is known from the data protection area as "short notices", "layered approach" and "privacy symbols". The transfer into the world of data, here the DSA, is done on that line, neglecting the possibility for a computer to determine the precise mismatch of preferences and policies, of contextual interaction or dynamic consent. The conclusions really merit to be extended by interdisciplinary thoughts and ideas. Unfortunately this wasn't done.

---

### Official Review · ~Kimberly_Garcia1 · 2025-07-21
**This paper describes some shortcomings of the Digital Services Act. In particular, the paper analysis the transparency paradox, a risk that companies might fall into while trying to comply with the DSA. This contribution argues that the current regulation guidelines are not enough to empower users, since they do not provide clear and standardized guidance on what and how to communicate with users of online services. The contribution calls policy makers to shift the formulation of regulation from a compliance perspective, to a more practical and structured one, in which transparency is meaningful, through contextual communication, user-tested interface design, and standardized reporting to actually provide information that can empower users of online services.**

**Rating:** 8
**Confidence:** 4

**Review:**

This is a well written paper that motivates the problem of current regulations, such as the DSA, it explores the risks that companies might incur in when only trying to comply with regulation, if such regulation does not provide clear guidance on how to provide users of online services with significant benefit. In the following, I provide a review by section and propose few restructuring changes to improve the flow of the contribution, and to increase the depth of the sections.

Section 2 introduces the problem, highlighting that mandatory disclosure is the most common and least effective form of regulation

Section 3 looks closer at the specifications of the DSA and compares it with privacy notices and the fact that even when regulations are in place, and companies seemingly comply with it, users who read these notices do not understand them either because of length or complexity in the terms used.

I think this section could be reinforced with an analysis on the GDPR and privacy notices, making the parallel with older, more established regulation could help to motivate the problem of a disclosure-only approach.

Section 4 is repetitive with section 3. Section 4 takes the study by Pasquale, F. (2020) to motivate the transparency paradox. However, this point is already made in section 3. Hence, consider merging the new analysis presented in 4 with section 3.

Section 5. This section introduces a paradigm called “Digital vulnerability”, calling for guidance and tools to create technology (e.g., applications and algorithms) that do not make users vulnerable by for example exploiting their data. This introduction of this paradigm is very interesting, I would like to see it better connected with the rest of the paper, perhaps by making a better connection to section 6.

Section 6
Takes an overview perspective to argue why transparency does not only reach platform operations, but it has a bigger societal impact, as most journalism is run over social media platforms, and algorithms shape the perceived reality of several groups, regarding politics, and even democratic processes. Again, this section is very interesting. However, I think it could be better embedded in the paper, by for example merging it with the content of section 5, into a section called “Digital Vulnerability and Algorithmic Transparency” or something such. Merging sections would improve the flow of the paper, and it would add depth to the narrative of the sections.

This is the same case for section 7 to 9. Section 8 and 9 could be weaved into 7, to consolidate the ideas, and avoid repetitions.

Other questions:
Even when overcoming the transparency paradox, what could be done to overcome phenomena such as the “privacy paradox"? in which even though users report that they care about their privacy, they are still willing to provide their data to get a service “for free” for example.
It would also be interesting to explore a technological perspective to this analysis. A lot of research on for example data privacy, privacy by design, dark patterns etc. exist to create tools and technological means to make people aware of the implications of for example accepting seamlessly free services. However, not many of the created tools are widely used, since public awareness and enforcement are not prioritised.

---

### Decision · Program_Chairs · 2025-07-25

Accept